# Identification *of SARS-CoV-2* and *Enteroviruses* in Sewage Water—A Pilot Study

**DOI:** 10.3390/v13050844

**Published:** 2021-05-06

**Authors:** Anda Băicuș, Carmen Maria Cherciu, Mihaela Lazăr

**Affiliations:** 1Department of Microbiology III—Emergency University Hospital Bucharest, Faculty of Medicine, “Carol Davila” University of Medicine and Pharmacy, 050098 Bucharest, Romania; 2Departament of Viral Infections, “Cantacuzino” Medico Military National Institute of Research and Development, 050096 Bucharest, Romania

**Keywords:** SARS-CoV-2, poliovirus, enterovirus, sewage water

## Abstract

Due to the outbreak of Severe Acute Respiratory Syndrome Coronavirus 2 (SARS-CoV-2), combined with the risk of polio importation from Ukraine, we evaluated the presence of SARS-CoV-2 and enteroviruses in 25 sewage water samples from Romania, concentrated using the WHO method between January 2020 and January 2021. Surveillance for enteroviruses and SARS-CoV-2 are relevant in the calculation of prevalence estimates as well as early detection of the introduction or disappearance of these viruses. For SARS-CoV-2 detection, we used two immunochromatographic nucleocapsid antigenic tests as well as real-time PCR assays, produced for respiratory samples. The isolation of cell culture lines, in accordance with the WHO recommendations, was carried out for enterovirus detection. Twenty-three of the samples investigated were positive in rapid tests for SARS-CoV-2, while the RNA of SARS-CoV-2, detected with Respiratory 2.1 plus a panel Biofire Film array, was present in eight samples. The Allplex 2019-nCoV assay was used for the validation of the tests. There were three genes detected in one sample, E, RdPR, and N, two genes, E and RdPR, in one sample, two genes, RdPR and N, in four samples, one gene, RdPR, in five samples and one gene, N, in one sample. Eight samples were positive for non-polio enteroviruses, and no poliovirus strains were isolated. This study suggests the presence of SARS-CoV-2 and enteroviruses in Romanian sewage water in 2020. As such, our results indicate that a rapid, more specific test should be developed especially for the detection of SARS-CoV-2 in sewage water.

## 1. Introduction

Launched in 1988, the Global Polio Eradication Initiative aims to eradicate poliovirus (PV) through extensive vaccination worldwide [1]. In June 2002, all 53 countries in the WHO European Region were certified as polio-free [2]. An outbreak due to the type 1 vaccine-derived poliovirus strain (VDPVs) took place in 2015 in Ukraine, a country that had been previously identified as at risk because of its low vaccination coverage [3]. Due to the risk of poliovirus importation, and the emergency of the vaccine-derived poliovirus (VDPV) strains, environmental enterovirus surveillance was enhanced, beginning in 2015 in the north and southeast of Romania, in the regions that border Ukraine. The sewage water concentration and poliovirus detection were performed according to the WHO guidelines for environmental polio surveillance [4]. Combined surveillance of enterovirus circulation in healthy children from at-risk areas and in the environment was evaluated in 2016. No poliovirus strains were isolated from 2009 to 2021 in Romania in the framework of acute flaccid paralysis or via the environmental surveillance systems. A high level of circulation of echovirus types 6 and 7 and coxsackievirus type B5 was recorded in 2016 [5,6,7].

In December 2019, the outbreak of a new betacoronavirus, named Severe Acute Respiratory Syndrome Coronavirus 2 (SARS-CoV-2), was first detected in Wuhan (China), which spread rapidly [8]

SARS-CoV-2 is an enveloped, positive-sense single-stranded RNA virus [9] with a genome size of approximatively 29.9 kb [10]. The genome sequence shares ~80% sequence identity with SARS-CoV and ~50% with MERS-CoV [11]. SARS-CoV-2 contains four structural proteins. The nucleocapsid protein (N) forms the capsid outside the genome, and the genome is also packed with an envelope that is associated with three structural proteins: a membrane protein (M), a spike protein (S), and an envelope protein (E). Respiratory transmission is the primary route of SARS-CoV-2 infection, but fecal–oral transmission is also possible as the virus can be detected in stool samples. In a study evaluating virus dynamics in Zhejiang Province in China, it was observed that the median duration of the existence of SARS-CoV-2 in stools was 22 days (range 17–31 days), which is longer than it remains in respiratory airways (18 days) [12].

Considering that SARS-CoV-2 and enteroviruses may survive for up to several days outside of the human body, their measurement in sewage water is relevant in the calculation of prevalence estimates as well as in the early detection of the introduction or disappearance of these viruses. However, the survival period of viruses in water environments depends on the temperature, the properties of the water, the concentration of suspended solids and organic matter, the solution pH, and the dose of disinfectant used [13].

We evaluated the co-circulation of SARS-CoV-2 between January 2020 and January 2021 alongside enteroviruses in samples collected monthly from the regions of Romania that border Ukraine.

## 2. Materials and Methods

Within the framework of environmental poliovirus surveillance, the Romanian public health authorities collected monthly samples by grab sampling sewage water (1000 mL/sample) from four different regions at the border with Ukraine (Figure 1) and sent them for virological investigations to the Enteric Viral Infections Laboratory, Cantacuzino Medico Military National Institute for Research and Development, Bucharest, Romania. In our pilot study, 25 sewage water samples were investigated, four collected in January 2020, 17 in the months of October, November, and December 2020, and four in January 2021. From each sample, 500 mL of sewage water was concentrated using the two-phase separation method and decontaminated by chloroform extraction, as recommended by the WHO guidelines [4]. Enteroviruses were detected by isolation on cell culture lines. L20B (a genetically engineered mouse cell line expressing the human poliovirus receptor PVR) and RD cells (derived from a rhabdomyosarcoma) were inoculated with the samples, as recommended by the WHO. RD cells are sensitive to most enteroviruses, and L20B is highly specific to poliovirus [14,15]. Positive samples on RD cell lines and negative samples on L20B cell lines were reported as non-polio enterovirus isolates (NPEV). Negative samples on RD cell lines and negative samples on L20B cell lines were reported as negative for poliovirus.

For the rapid detection of SARS-CoV-2, the sewage water samples were tested with two rapid immunochromatographic nucleocapsid antigenic tests, Biotech Laboratories’ NG Test SARS-CoV-2 Ag, and the Siemens Clinitest Rapid COVID-19 antigen test, which use highly sensitive monoclonal antibodies to detect the nucleocapsid protein of SARS-CoV-2 in nasopharyngeal samples. The tests have a control that indicates that a proper volume of sample has been added. For the NG Test SARS-CoV-2 Ag, the manufacturer claims a sensitivity of 92% [95% CI: 80–97%], and a specificity of 100% [95%CI: 96–100%] for nasopharyngeal samples. For the Siemens Clinitest Rapid COVID-19 antigen test, the limit of detection (LOD) for SARS-CoV-2 is 1.15 × 10^2^ TCID_50_/mL, while the sensitivity is 96.72% [95% CI: 88, 65–96.60%] and the specificity is 99.22% [95% CI: 96–100%].

We adapted the protocols for the sewage water samples. For the NG Test SARS-CoV-2 Ag, we dispensed four droplets of buffer (100 µL), 100 µL of sewage water diluted 1:4 (100 µL of concentrated sewage water with 300 µL of Eagle’s Minimum Essential Medium) and, after that, we followed the steps recommended by the manufacturer. For the Siemens Clinitest Rapid COVID-19 antigen test, we dispensed 10 droplets of buffer (300 µL), 100 µL of sewage water concentrated using the WHO method and, after that, we followed the steps recommended by the manufacturer.

For the molecular detection of SARS-CoV-2, 300 µL of concentrated sewage water was tested with the Respiratory 2.1 plus panel Biofire Film array, a multiplex PCR system that integrates sample preparation, amplification, detection, and analysis into one system that requires a total runtime of about an hour. The panel tests for 19 viruses and four bacteria that cause respiratory tract infections, with an overall sensitivity and specificity of 97.4% and 99.4%, respectively, for the nasopharyngeal samples. The pathogens detected are Adenovirus, Coronavirus 229E, Coronavirus HKU1, Coronavirus OC43, Coronavirus NL63, Middle East Respiratory Syndrome Corona Virus (MERS-CoV), Severe Acute Respiratory Syndrome Coronavirus 2 (SARS-CoV-2), Human Metapneumovirus, Human Rhinovirus/Enterovirus, Influenza A, Influenza A/H1, Influenza A/H1-2009, Influenza A/H3, Influenza B, Parainfluenza 1, Parainfluenza 2, Parainfluenza 3, Parainfluenza 4, RSV, *Bordetella pertussis*, *Bordetella parapertussis*, *Chlamydophila pneumoniae*, and *Mycoplasma pneumoniae*. The panel does not differentiate between Rhinovirus and Enterovirus as they belong to the same family, *Picornaviridae*.

The Allplex 2019-nCoV assay, Seegene, was used for the validation of the results obtained and for the qualitative detection of SARS-CoV-2 with real-time reverse transcription PCR after the extraction of the RNA from the sewage water with King Fisher Flex 96 equipment, in accordance with the manufacturer’s protocol.

The presence of SARS-CoV-2 was evaluated as a function of the presence of the E, RdPR (RNA-dependent RNA polymerase), and N genes. The detection of the E, RdPR, and N genes in the samples was interpretated as the detection of SARS-CoV-2. When only two genes were detected, a re-test was recommended with an increased sample concentration, while the detection of only E was interpretated as a negative result for the presence of SARS-CoV-2.

## 3. Results

Of the 25 sewage water samples inoculated on cell culture lines, eight (8%) were positive for non-polio enteroviruses. No poliovirus strains were isolated.

Twenty-three out of 25 sewage water samples investigated with the rapid tests NG Test SARS-CoV-2 Ag and the Siemens Clinitest Rapid COVID-19 antigen test were positive for SARS-CoV-2. The molecular investigations with the Respiratory 2.1 panel plus Biofire Film array detected SARS-CoV-2 in eight samples, Human Rhinovirus/Enterovirus in 22 samples, and Adenovirus in 23 samples. With the Allplex 2019-nCoV assay, all three SARS-CoV-2 genes (E, RdPR, and N) were detected in one sample collected from Siret. Two genes, E and RdPR, were detected in one sample, two genes, RdPR and N, were detected in four samples, the RdPR gene was detected in five samples, and the N gene was detected in one sample (Table 1).

In Babadag, the sample collected in October 2020 contained SARS-CoV-2, but the virus was not detected in the sample collected in January 2021. In the sample collected from Sighetu Marmatiei in January 2020, Coronavirus HKU 1 was detected. In November and December 2020, SARS-CoV-2 was detected; however, in January 2021, no coronaviruses were detected. At two sites (Viseu and Borsa), SARS-CoV-2 was absent in the October samples, identified in the November and December 2020 samples, and absent in January 2021 in Viseu.

Adenoviruses are often considered the most abundant human viral pathogens in sewage water, and in a risk assessment of exposure to biosolid-derived aerosols, adenoviruses accounted for a significantly higher predicted risk than enteroviruses [16,17]. In our study, *Adenovirus* was detected in 92% of samples and non-polio enteroviruses in 32% of samples (Table 2).

## 4. Discussion

The Global Polio Eradication Initiative has succeeded in eradicating two serotypes of wild poliovirus, types 2 and 3. In 2021, wild poliovirus type 1 currently affects two countries: Pakistan and Afghanistan. In Romania, poliomyelitis was controlled by using a trivalent oral polio vaccine (tOPV) until 2008, and vaccination with the inactivated polio vaccine (IPV) started in 2009. Due to the risk of polio importation and the low level of polio vaccine coverage in different areas, the national authorities of public health decided to enhance environmental polio surveillance in Romania in 2015. In total, 1211 sewage water samples were collected from at-risk areas and virologically investigated at the Enteric Viral Infections Laboratory, Cantacuzino Medico Military National Institute for Research and Development, Bucharest, Romania, between 2015 and 2019. Four hundred and twenty-eight (35%) non-polio enterovirus strains were isolated.

No poliovirus strains were isolated from 2009 to 2021 in Romania in the framework of the surveillance of acute flaccid paralysis (AFP) cases and the environmental surveillance program.

At the beginning of 2020, we started a prospective study on sewage water samples concerning the co-circulation of respiratory pathogens with enteroviruses. We used the Respiratory 2.1 panel Biofire Film array, which could not detect SARS-CoV-2 at that time. We detected the same respiratory pathogens, but, of course, they were detected without SARS-CoV-2. In the present study, 23 sewage water samples were positive in rapid tests for SARS-CoV-2, meaning either that the antigenical structure was present in all samples, or that the rapid antigenic SARS-CoV-2 tests were cross-reactive with other proteic structures present in sewage water (as the gene N of SARS-CoV-2 was detected in six (26%) samples), while some results could be considered as false positives. The latter explanation is even more likely when considering the positivity rate, and the fact that the test has no antigenic control. From eight samples positive for SARS-CoV-2 detected with the Biofire Film array, three genes were detected in one sample (E, RdPR, and N), two genes were detected in five samples, while no genes were detected in two samples, so we can consider that the Biofire Film array has a high sensitivity for virus detection in sewage water.

The co-circulation of Human Rhinovirus/Enterovirus, Adenovirus, and SARS-CoV-2 was recorded by molecular detection in sewage water in Romania between October and December 2020. In relation to the WHO polio laboratory network, these results are important and may help to update the safety measures regarding the concentration of sewage water samples.

Taking into account our results, we consider that, for the detection of SARS-CoV-2 in sewage waters, an increase in the sample concentration and the use of the Biofire Film array could be applied in an emergency situation.

## Figures and Tables

**Figure 1 viruses-13-00844-f001:**
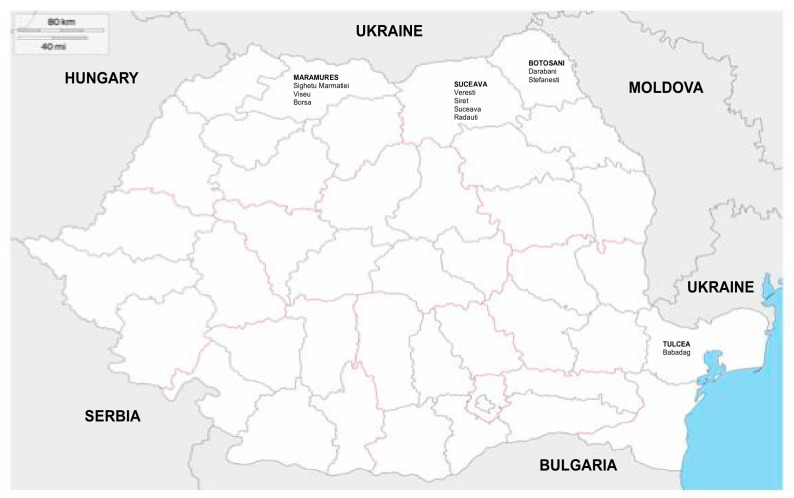
Map of Romania. The locations of environmental sampling.

**Table 1 viruses-13-00844-t001:** Epidemiological data and the results of the virological and molecular investigations.

No.	Region	Collection Site and Date–Laboratory Arrival Date	NG Test SARS-CoV-2 Ag	Siemens Clinitest Rapid COVID-19 Antigen Test	Respiratory 2.1 Panel Plus Biofire FILM Array	Enterovirus Isolation on RD and L20B Cell Culture Lines	*E* ** *Gene*	*RdPR* ** *Gene*	*N* ** *Gene*
1	SV	VERESTI 4.11–5.11.2020	Positive	Positive	*Adenovirus*	Negative	ND	D	D
2		SIRET 4.11–5.11.2020	Positive	Positive	*Adenovirus* *Human Rhinovirus/EV ** *SARS-CoV-2*	Negative	D	D	D
3		SUCEAVA 4.11–5.11.2020	Positive	Positive	*Adenovirus, Human Rhinovirus/EV*	Negative	ND	ND	ND
4		RADAUTI 4.11–5.11.2020	Positive	Positive	*Negative*	Negative	ND	ND	D
5	BT	DARABANI 25.08–26.08.2020	Positive	Positive	*Adenovirus* *Human Rhinovirus/EV*	*Non-polio enterovirus*	ND	ND	ND
6		DARABANI 3.11–4.11.2020	Positive	Positive	*Adenovirus* *Human Rhinovirus/EV*	*Non-polio enterovirus*	ND	D	ND
7		DARABANI 16.02–17.02.2021	Negative	Positive	*Adenovirus* *Human Rhinovirus/EV*	Negative	ND	ND	ND
8		STEFANESTI 3.11–4.11.2020	Positive	Positive	*Adenovirus* *Human Rhinovirus/EV*	*Non-polio enterovirus*	ND	ND	ND
9		STEFANESTI 25.08–26.08.2020	Positive	Positive	*Negative*	Negative	ND	D	D
10		BABADAG 15.01–16.01.2020	Positive	Negative	*Adenovirus* *Human Rhinovirus/EV*	Negative	ND	ND	ND
11	TL	BABADAG 28.10–29.10.2020	Positive	Positive	*Adenovirus* *Human Rhinovirus/EV* *SARS-CoV-2*	Negative	ND	ND	ND
12		BABADAG 20.01–25.01.2021	Negative	Positive	*Adenovirus* *Human Rhinovirus/EV*	Negative	ND	ND	ND
13		SIGHETUL MARMATIEI 14.01–15.01.2020	Positive	Negative	*Adenovirus* *Human Rhinovirus/EV* *Coronavirus HKU1*	*Non-polio enterovirus*	ND	ND	ND
14	MM	SIGHETUL MARMATIEI 10.11–11.11.2020	Positive	Positive	*Adenovirus* *Human Rhinovirus/EV* *SARS-CoV-2*	Negative	D	D	ND
15		SIGHETUL MARMATIEI 8.12–9.12.2020	Positive	Positive	*Adenovirus,* *Human Rhinovirus/EV* *SARS-CoV-2*	Negative	ND	D	D
16		SIGHETUL MARMATIEI 12.01–13.01.2021	Positive	Positive	*Adenovirus,* *Human Rhinovirus/EV*	Negative	ND	ND	ND
17		VISEU 14.01–15.01.2020	Positive	Positive	*Adenovirus, Human Rhinovirus/EV*	Negative	ND	ND	ND
18		VISEU 13.10–14.10.2020	Positive	Positive	*Adenovirus, Human Rhinovirus/EV*	*Non-polio enterovirus*	ND	D	ND
19		VISEU 10.11–11.11.2020	Positive	Positive	*Adenovirus* *Human Rhinovirus/EV* *SARS-CoV-2*	Negative	ND	D	ND
20		VISEU 8.12–9.12.2020	Positive	Positive	*Adenovirus* *Human Rhinovirus/EV* *SARS-CoV-2*	*Non-polio enterovirus*	ND	ND	ND
21		VISEU 12.01–13.01.2021	Positive	Positive	*Adenovirus* *Human Rhinovirus/EV*	Negative	ND	ND	ND
22		BORSA 14.01–15.01.2020	Positive	Positive	*Adenovirus* *Human Rhinovirus/EV*	Negative	ND	ND	ND
23		BORSA 13.10–14.10.2020	Positive	Positive	*Adenovirus* *Human Rhinovirus/EV*	*Non-polio enterovirus*	ND	D	ND
24		BORSA 10.11–11.11.2020	Positive	Positive	*Adenovirus* *Human Rhinovirus/EV* *SARS-CoV-2*	Negative	ND	D	ND
25		BORSA 8.12–9.12.2020	Positive	Positive	*Adenovirus, Human Rhinovirus/EV* *SARS-CoV-2*	*Non-polio enterovirus*	ND	D	D

* EV = enterovirus; ** Allplex 2019 nCov RT-PCR assay; ND—not detected; D—detected. Region: SV—Suceava, BT—Botosani, TL—Tulcea, MM—Maramures.

**Table 2 viruses-13-00844-t002:** January 2020–January 2021 timeline (samples, tests, and results).

Number samples	27	77	26	8
Virus searched for	Polio surveillance Biofire respiratory panel+SARS-CoV-2	Polio surveillance Biofire respiratory panel (detecting MERS and 4 coronavirus strains)	Polio surveillance Biofire respiratory panel+SARS-CoV-2	Polio surveillance Biofire respiratory panel+SARS-CoV-2
Virus found	8 NPEV/27 samples 4 Adenovirus/ 4 samples 4 Human Rhinovirus/Enterovirus/4 samples	7 NPEV */77 samples 34 Adenovirus/53 samples 17 Human Rhinovirus/Enterovirus/53 samples	10 NPEV/26 samples 15 Adenovirus/17samples 14 Human Rhinovirus/Enterovirus/ 17 samples 8 SARS-CoV-2/17 samples	2 NPEV//8 samples 4 Adenovirus/ 4 samples 4 Human Rhinovirus/Enterovirus/4 samples
Period	January–February **	March–September	October–December	January 2021

* Non-polio enterovirus detected by isolation of cell culture lines; **26/02/2020: first case of diagnosed COVID-19 in Romania.

## Data Availability

Data is contained within this article.

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
