# Peer review of "Identification of SARS-CoV-2 and Enteroviruses in Sewage Water—A Pilot Study"

_viruses, 2021, doi:10.3390/v13050844_

Round 1

Reviewer 1 Report

The manuscript has improved. However table 2 should be more the focus of the results than table 1. It really nicely shows the detction of SARS-CoV and EVs as well as other viruses which gives the impact of incorporating a good virus detection package on environmental surveillance especially during a pandemic. However the clinical data in the area is still missing the importance of this algoritm of the environmental. A simple addition to table 2 (which should be table 1) of a row stating teh number of SARS-Cov-2 and EV clinical cases in the area would be really heplfull.

tabel 1 is more of a detail of table 2 showing the samples testen for SARS additionally with rapid antigen tests which is nicely discussed allready on the difference between the microarray and antigen test. 

in table 1 column " country" should be "region" and should be explained in the notes what the abbreviations are.

Author Response

Cover letter

We thank very much to the reviewers for their time and effort spent to make our article better.

We made our best to respond to all their suggestions.

1.Comments and Suggestions for Authors

The manuscript has improved. However table 2 should be more the focus of the results than table 1. It really nicely shows the detction of SARS-CoV and EVs as well as other viruses which gives the impact of incorporating a good virus detection package on environmental surveillance especially during a pandemic. However the clinical data in the area is still missing the importance of this algoritm of the environmental. A simple addition to table 2 (which should be table 1) of a row stating teh number of SARS-Cov-2 and EV clinical cases in the area would be really heplfull.

tabel 1 is more of a detail of table 2 showing the samples testen for SARS additionally with rapid antigen tests which is nicely discussed allready on the difference between the microarray and antigen test. 

in table 1 column " country" should be "region" and should be explained in the notes what the abbreviations are.

  • We did not have access to the number of SARS-Cov-2 and EV clinical cases in the area, there are no published data on counties.

Abstract

We  started this study supposing that the SARS CoV-2 could be isolated from faeces during the mid to late clinical phase of symptoms.

Line 12. It was introduced the sentence: Considering that SARS -CoV-2 and enteroviruses may survive for up to several days out of the human body, its measurement in sewage water could provide a tool for risk assessment across communities.  

Introduction

Line 30-31 It was introduced- and the emergency of the vaccine derived poliovirus (VDPV) strains

Line 32  It was introduced- and the south-east of………… regions

Line 34  It was introduced- reference  [4]

Line 36 It was deleted ref [4,5]

Line  39  references [5-7] have been introduced

Line 42  reference [7] became  the reference [8]

Line 43 reference [8]   became the reference [9]

Line 44 reference [9] became the reference [10]

Line 45 reference [10] became the reference [11]

Line 53 reference [11]  became  the reference [12]

Line 58 reference [12]  became the reference [13]

Line 60 – It was changed the word “districts” with regions

Materials and Methods

Line 62 -  It was changed the word “district” with “regions”

Line 64 – It was introduced (Fig. 1.)

Line 70 - reference [13] became  the reference [4]

Line 74 – “RD cells can be infected by” became  “RD cells is sensitive to”; and “ L20B can be infected only by poliovirus” became” L20B  is highly specific for poliovirus”

Line 119 – It was introduced the Figure 1. Map of Romania. The locations of environmental sampling

Results

Table 1 It was changed the word “county”  with “region”, the abbreviations were explained  

SV- Suceava, BT-Botosani, TL–Tulcea , MM-Maramures 

Discussion

Line 144 The information about the Global Polio Eradication Initiative and the polio surveillance in Romania were introduced.

The Global Polio Eradication Initiative has succeeded in eradicating two serotypes of wild poliovirus, types 2 and 3. In 2021, wild poliovirus type 1 affects two countries: Pakistan and Afghanistan. In Romania, poliomyelitis was controlled by using a trivalent oral polio vaccine (tOPV) until 2008, and vaccination with the inactivated polio vaccine (IPV) started in 2009. Due to the risk of polio importation and the low level of polio vaccine coverage in different areas, the national authorities of public health decided to enhance the polio environmental surveillance in Romania since 2015. 1211 sewage water samples were collected from at risk areas and virologically investigated at the Enteric Viral Infections Laboratory, Cantacuzino Medico-Military National Institute for Research and Development, Bucharest, Romania between 2015-2019.  428 (35%) non polio enterovirus strains were isolated

It was deleted “In the framework of the poliovirus environment surveillance we investigated” and the sentence started with  “ Between January 2020 - January 2021, we investigated 138 sewage water samples collected from the northern and the south-east of Romania, at the border with Ukraine. 27 ( 19,5%) non polio enterovirus strains were isolated on cell culture lines, (Table 2).

It was introduces the sentence “No poliovirus strains were isolated from 2009 to 2021 in Romania in the framework of the surveillance of the acute flaccid paralysis (AFP) cases, and of the environmental surveillance”.

References

We inserted the requested information in the ref 1,4,14,15.

  1. Kew, O.; Pallansch, M. Breaking the Last Chains of Poliovirus Transmission: Progress and Challenges in Global Polio Eradication. Rev. Virol, 2018 Sep 29; 5(1), pp. 427–451; 10.1146/annurev-virology-101416-041749.
  2. World Health Organization . (2003) Guidelines for environmental surveillance of poliovirus circulation. World Health Organization .https://apps.who.int/iris/handle/10665/67854.
  3. World Health Organization. (‎2004)‎. Polio laboratory manual, 4th ed. World Health      https://apps.who.int/iris/handle/10665/68762.
  4. World Health Organisation. 2017. WHO Polio Laboratory Manual/Supplement 1: An alternative test algorithm for poliovirus isolation and characterization. Available at http://polioeradication.org/tools-and-library/policy-reports/gpln-publications/. Accessed 4 Jan 2021.

Reviewer 2 Report

Abstract. The purpose of the study should be formulated more clearly. What was the purpose of the SARS-CoV-2 detection in wastewater samples? This is understandable for poliovirus. 

Line 28-32. Romania, like Ukraine, belongs to the countries of risk (https://www.euro.who.int/en/health-topics/communicable-diseases/poliomyelitis/publications/2019/33rd-meeting-of-the-european-regional-commission-for-certification-of-poliomyelitis-eradication-rcc-report-2019). Therefore, the purpose of the study of wastewater should be formulated more correctly, for example, due to the risk of VDPV formation, not only "due to the risk of poliovirus importation".

Line 27, ref [1]. The link is not entirely correct. It is recommended to refer to the original document of the WHO. 

Lines 32-34. A link to the referenced WHO guidelines is needed. 

Lines 72-74. A link to WHO Manual is needed.

Line 74. "RD cells can be infected by most enteroviruses, and L20B can be infected ..." - not infected, but sensitive. 

Materials and Methods. It is recommended to provide a map showing the locations of environmental sampling. 

Discussion. Environmental surveillance was organized in Romania in the context of the global polio eradication program. The authors write about this in the introduction. As a result of the presented study, no poliovirus was found in wastewater. This fact should be commented on in the discussion. Authors should also specify the type of poliovirus vaccine used in Romania.

Author Response

Cover letter

We thank very much to the reviewers for their time and effort spent to make our article better.

We made our best to respond to all their suggestions.

  1. Comments and Suggestions for Authors

Abstract. The purpose of the study should be formulated more clearly. What was the purpose of the SARS-CoV-2 detection in wastewater samples? This is understandable for poliovirus. 

Abstract

We  started this study supposing that the SARS CoV-2 could be isolated from faeces during the mid to late clinical phase of symptoms.

Line 12. It was introduced the sentence: Considering that SARS -CoV-2 and enteroviruses may survive for up to several days out of the human body, its measurement in sewage water could provide a tool for risk assessment across communities.  

Line 27, ref [1]. The link is not entirely correct. It is recommended to refer to the original document of the WHO.  

  • We inserted the requested information in the ref [1]

Kew O, Pallansch M. Breaking the Last Chains of Poliovirus Transmission: Progress and Challenges in Global Polio Eradication. Annu Rev Virol. 2018 Sep 29;5(1):427-451. doi: 10.1146/annurev-virology-101416-041749.

Line 28-32. Romania, like Ukraine, belongs to the countries of risk (https://www.euro.who.int/en/health-topics/communicablediseases/poliomyelitis/publications/2019/33rd-meeting-of-the-european-regional-commission-for-certification-of-poliomyelitis-eradication-rcc-report-2019). Therefore, the purpose of the study of wastewater should be formulated more correctly, for example, due to the risk of VDPV formation, not only "due to the risk of poliovirus importation". 

Line 30-31 It  was introduced- and the emergency of the vaccine derived poliovirus (VDPV) strains

Lines 32-34. A link to the referenced WHO guidelines is needed. 

  • We inserted the requested information in the ref [4]

World Health Organization. (‎2003)‎. Guidelines for environmental surveillance of poliovirus circulation. World Health Organization. https://apps.who.int/iris/handle/10665/67854

Line 32:  It was introduced- and the south-east of ………… regions

Line 34:  It was introduced- reference  [4]

Line 36: It was deleted ref [4,5]

Line  39  references [5-7] have been introduced

Line 42  reference [7] became  the reference [8]

Line 43 reference [8]   became the reference [9]

Line 44 reference [9] became the reference [10]

Line 45 reference [10] became the reference [11]

Line 53 reference [11]  became  the reference [12]

Line 58 reference [12]  became the reference [13]

Line 60 – It was changed the word “districts” with regions

Materials and Methods

Line 62 -  It was changed word “district” with “regions”

Line 64 – It was introduced (Fig. 1.)

Line 70 - reference [13] became  the reference [4]

Lines 72-74. A link to WHO Manual is needed.

  • We inserted the requested information in the ref [14, 15]

  • World Health Organization.(‎2004)‎. Polio laboratory manual, 4th ed. World Health Organization. https://apps.who.int/iris/handle/10665/68762.
  1. World Health Organisation. 2017. WHO Polio Laboratory Manual/Supplement 1: An alternative test algorithm for poliovirus isolation and characterization. Available at http://polioeradication.org/tools-and-library/policy-reports/gpln-publications/. Accessed 4 Jan 2021.

Line 74. "RD cells can be infected by most enteroviruses, and L20B can be infected ..." - not infected, but sensitive. 

Line 74 – “RD cells can be infected by” became  “RD cells is sensitive to”; and “ L20B can be infected only by poliovirus” became” L20B  is highly specific for poliovirus”

Materials and Methods. It is recommended to provide a map showing the locations of environmental sampling. 

Line 119 – It was introduced Figure 1. Map of Romania. The locations of environmental sampling

Results

Table 1 It was changed the word “county”  withregion”, the abbreviations were explained  

SV- Suceava, BT -Botosani,TL – Tulcea , MM- Maramures 

Discussion. Environmental surveillance was organized in Romania in the context of the global polio eradication program. The authors write about this in the introduction. As a result of the presented study, no poliovirus was found in wastewater. This fact should be commented on in the discussion. Authors should also specify the type of poliovirus vaccine used in Romania.

Line 144 The information about the Global Polio Eradication Initiative and the polio surveillance in Romania were introduced.

The Global Polio Eradication Initiative has succeeded in eradicating two serotypes of wild poliovirus, types 2 and 3. In 2021, wild poliovirus type 1 affects two countries: Pakistan and Afghanistan. In Romania, poliomyelitis was controlled by using a trivalent oral polio vaccine (tOPV) until 2008, and vaccination with the inactivated polio vaccine (IPV) started in 2009. Due to the risk of polio importation and the low level of polio vaccine coverage in different areas, the national authorities of public health decided to enhance the polio environmental surveillance in Romania since 2015. 1211 sewage water samples were collected from at risk areas and virologically investigated at the Enteric Viral Infections Laboratory, Cantacuzino Medico-Military National Institute for Research and Development, Bucharest, Romania between 2015-2019.  428 (35%) non polio enterovirus strains were isolated

It was deleted “In the framework of the poliovirus environment surveillance we investigated” and the sentence started with  “ Between January 2020 - January 2021, we investigated 138 sewage water samples collected from the northern and the south-east of Romania, at the border with Ukraine. 27 ( 19,5%) non polio enterovirus strains were isolated on cell culture lines, (Table 2).

It was introduces the sentence “No poliovirus strains were isolated from 2009 to 2021 in Romania in the framework of the surveillance of the acute flaccid paralysis (AFP) cases, and of the environmental surveillance”.

References

We inserted the requested information in the ref 1,4,14,15.

  1. Kew, O.; Pallansch, M. Breaking the Last Chains of Poliovirus Transmission: Progress and Challenges in Global Polio Eradication. Rev. Virol, 2018 Sep 29; 5(1), pp. 427–451; 10.1146/annurev-virology-101416-041749.
  2. World Health Organization . (2003) Guidelines for environmental surveillance of poliovirus circulation. World Health Organization .https://apps.who.int/iris/handle/10665/67854.
  3. World Health Organization. (‎2004)‎. Polio laboratory manual, 4th ed. World Health      https://apps.who.int/iris/handle/10665/68762.
  4. World Health Organisation. 2017. WHO Polio Laboratory Manual/Supplement 1: An alternative test algorithm for poliovirus isolation and characterization. Available at http://polioeradication.org/tools-and-library/policy-reports/gpln-publications/. Accessed 4 Jan 2021.

This manuscript is a resubmission of an earlier submission. The following is a list of the peer review reports and author responses from that submission.

Round 1

Reviewer 1 Report

The author describes the detection of enteroviruses (EVs) and SARS-CoV-2 from sewage in Romania. While this co-detection is of interest, the manuscript would benefit more, by including data from previous years were these was no SARS-CoV-2 circulation, as well as the clinical circulation of EVs and SARS-CoV-2 in the area. This will put the data described in perspective of low or high circulation of these two viruses during the study period and in comparison to previous years on how this algorithm is of added value. The manuscript should be read by a native English speaker as some sentences are unclear.

Specific comments

  • material methods: please describe the collection frequency of sewage sampling in the area.
  • Simens should be Siemens.
  • What is the LLOD of the (adapted) Antigen protocols
  • What is the LLOD of the molecular assay. Otherwise indicate a reference.
  • It has been suggested that Antigen tests can bring forth false positive results. this can be addressed by analyzing (SARS-CoV2 negative samples from previous years when we know there was no SARS-CoV-2 circulation
  • Include in table 1 data from the antigen tests. In the table it seems you only found 8 pos SARS-CoV-2 yet in the text all are positive. Reading further along it seems that all are Antigen positive but only half are PCr positive. this needs to be described more clearly in the results section/table 1.
  • Conclusion: please provide more background on to why the author concludes that the presence of two positive strand ssRNA viruses in the same ecosystem could create conditions of a modified virus. Recombination is highly unlikely as recombination between two different virus families is unlikely. 

Author Response

The author describes the detection of enteroviruses (EVs) and SARS-CoV-2 from sewage in Romania. While this co-detection is of interest, the manuscript would benefit more, by including data from previous years were these was no SARS-CoV-2 circulation, as well as the clinical circulation of EVs and SARS-CoV-2 in the area. This will put the data described in perspective of low or high circulation of these two viruses during the study period and in comparison to previous years on how this algorithm is of added value. The manuscript should be read by a native English speaker as some sentences are unclear.

Concerning the clinical circulation of EVs in the area, there is reference 6 (Băicuș, A.; Joffret,  M.L.; Bessaud, M.; Delpeyroux, F.; Oprisan, G. Reinforced poliovirus and enterovirus surveillance in Romania, 2015-2016. Arch. Virol, 2020; 165(11), pp. 2627-2632). Stool samples of 155 healthy children and 186 samples of sewage waters were investigated then.

We re-wrote the entirely the Introduction and Discussion sections, and the English was corrected along the article. 

Specific comments

  • material methods: please describe the collection frequency of sewage sampling in the area.

The frequency was every month (we added this)

  • Simens should be Siemens.

Corrected

  • What is the LLOD of the (adapted) Antigen protocols
  • What is the LLOD of the molecular assay. Otherwise indicate a reference.

We inserted the requested information in the text

  • It has been suggested that Antigen tests can bring forth false positive results. this can be addressed by analyzing (SARS-CoV2 negative samples from previous years when we know there was no SARS-CoV-2 circulation

We will test samples older than 2020 in order to see if cross-reactivity was really an issue.

  • Include in table 1 data from the antigen tests. In the table it seems you only found 8 pos SARS-CoV-2 yet in the text all are positive. Reading further along it seems that all are Antigen positive but only half are PCr positive. this needs to be described more clearly in the results section/table 1.

We hope all is clear now, both in the text and table.

  • Conclusion: please provide more background on to why the author concludes that the presence of two positive strand ssRNA viruses in the same ecosystem could create conditions of a modified virus. Recombination is highly unlikely as recombination between two different virus families is unlikely. 

We deleted the affirmation. The long time multiplication of the viruses in intestine could create conditions for the emergence of modified viruses (by mutations in the genome) and not by recombination.  

Reviewer 2 Report

Overview: The article describes a yearlong study where the authors sampled for polio and other enteroviruses using a two phase concentration method. The authors used this same concentrate and tested for SARS2, using two antigen tests and a viral RT-PCR panel (BioFire Film Array). The authors did not find any poliovirus, but detected other enteroviruses in 44% of the samples. All the samples were positive for COVID antigen, but only 50% of samples were positive by the BioFire Film Array. The work itself is interesting but suffers from some limitations. 1) There are no controls for the antigen tests. The sewage concentrates are complex and have high potential of cross-reactivity with whatever is in the sewage. It would have been good to see other concentrates or samples before SARS2 was circulating to show that cross-reactivity is not an issue. The high number of positivity makes cross-reactivity the most likely explanation. The BioFire results are interesting. It was not clear that the discrepancy between the polio results and BioFire is due to probe. I’m guessing because the BioFIre is respiratory and polio procedure is enteric procedures the difference. That was also not clear.

Other concerns: There is significant wording and readability issues throughout the article. The introduction is especially problematic. The M&M is ok, so is the Results. The Conclusion needs work too.

Other comments:

Line 15 -16, 61 -63, 132-133: the authors allude to a new variant could arise from a mixture of +strand RNA viruses in sewage, this type of hyperbole is not needed.

Author Response

Overview: The article describes a yearlong study where the authors sampled for polio and other enteroviruses using a two phase concentration method. The authors used this same concentrate and tested for SARS2, using two antigen tests and a viral RT-PCR panel (BioFire Film Array). The authors did not find any poliovirus, but detected other enteroviruses in 44% of the samples. All the samples were positive for COVID antigen, but only 50% of samples were positive by the BioFire Film Array. The work itself is interesting but suffers from some limitations. 1) There are no controls for the antigen tests. The sewage concentrates are complex and have high potential of cross-reactivity with whatever is in the sewage. It would have been good to see other concentrates or samples before SARS2 was circulating to show that cross-reactivity is not an issue. The high number of positivity makes cross-reactivity the most likely explanation. The BioFire results are interesting. It was not clear that the discrepancy between the polio results and BioFire is due to probe. I’m guessing because the BioFIre is respiratory and polio procedure is enteric procedures the difference. That was also not clear.

We acknowledged the limitations in the Discussion section. We will test samples older than 2020 in order to see if cross-reactivity was really an issue.

Other concerns: There is significant wording and readability issues throughout the article. The introduction is especially problematic. The M&M is ok, so is the Results. The Conclusion needs work too.

We re-wrote the entirely the Introduction and Discussion sections. 

Other comments:

Line 15 -16, 61 -63, 132-133: the authors allude to a new variant could arise from a mixture of +strand RNA viruses in sewage, this type of hyperbole is not needed.

We deleted the affirmation.

Round 2

Reviewer 1 Report

The authors have addressed the comments adequately and the manuscript has improved considerable. However I have a few comments  on the details of the data collected at the beginning of 2020. It is presented in the discussion, but would be better also placed in the results, even though Antigen tests were not done and the array did not test for SARS-Cov-2.  As it will give a time line of the data. Also, samples were tested at the beginning of 2020, and the data in the table represent sewage samples at the end of 2020. What happened to the samples between? A graph on the detection of viruses based on microarray/culture from whole of 2020 from sewage would place the data in a temporal context as implied in the discussion that the virus could also be represent in the previous samples.

In that respect data on the clinical surveillanc eof SARS-CoV-2 and EV should be included in a graph in order to place the sewage data to to pandemic and EV circulation (or lack of)

Even better would be that all sewage samples would be retested with the new filmarray that includes sarscov-2.

Author Response

We thank very much to the reviewers for their time and effort spent to make our article better.

We made our best to respond to all their suggestions.

R 1 Comments and Suggestions for Authors

The authors have addressed the comments adequately and the manuscript has improved considerable. However I have a few comments  on the details of the data collected at the beginning of 2020. It is presented in the discussion, but would be better also placed in the results, even though Antigen tests were not done and the array did not test for SARS-Cov-2.  As it will give a time line of the data. Also, samples were tested at the beginning of 2020, and the data in the table represent sewage samples at the end of 2020. What happened to the samples between? A graph on the detection of viruses based on microarray/culture from whole of 2020 from sewage would place the data in a temporal context as implied in the discussion that the virus could also be represent in the previous samples.

We detailed the timeline concerning the detection of all searched viruses for the whole year 2020 in Table 2.

We think that the place of the information concerning the other viruses is in the Discussion section, as this was not among the objective of the present study.

In that respect data on the clinical surveillance of SARS-CoV-2 and EV should be included in a graph in order to place the sewage data to to pandemic and EV circulation (or lack of)

Even better would be that all sewage samples would be retested with the new filmarray that includes sarscov-2.

We will retest them with Biofire film array as soon as we will get financing.

Reviewer 2 Report

The article is a brief study on detecting COVID19 in sewage using the polio environmental surveillance network. The study is interesting but has few limitations that need to be clearly stated. The procedures and methods used were for respiratory samples not for environmental concentrates. Because the procedures and processes were adapted many of sensitivities and specificities would not be accurate. Nevertheless, the article would find interest to other public health scientist to determine what methods can be used for the detection of COVID19.

Specific issues:

Title: The title is not very accurate. Circulation in this instance is not correct. Maybe “Presence of …”or “ “Identification of SARS-COV-2…and Enteroviruses in Sewage Water” would be more accurate.

Line 14: “All samples…”. The authors must be clear that the 100% rapid was probably due to cross-reactivity. These results are false positives.

Line 86 -89: It is not clear what is meant by “100 ul of sewage water concentrated with 300 ul…”

Line 96-98: two points here. 1) Why are these words bolded? 2) Here is the issue previously brought up. Sensitivity and specificity is for respiratory samples not environmental samples from two phase. These numbers are misleading.

Discussion: Limitations are fine, but I think the authors should try to emphasize the false positives in the antigen tests. This is a serious limitation and authors should be very forthcoming with this information. Researchers would want to know that antigen tests for sewage samples are not a good test strategy.

Author Response

We thank very much to the reviewers for their time and effort spent to make our article better.

We made our best to respond to all their suggestions.

R2 Comments and Suggestions for Authors

The article is a brief study on detecting COVID19 in sewage using the polio environmental surveillance network. The study is interesting but has few limitations that need to be clearly stated. The procedures and methods used were for respiratory samples not for environmental concentrates. Because the procedures and processes were adapted many of sensitivities and specificities would not be accurate. Nevertheless, the article would find interest to other public health scientist to determine what methods can be used for the detection of COVID19.

Specific issues:

 Title: The title is not very accurate. Circulation in this instance is not correct. Maybe “Presence of …”or “ “Identification of SARS-COV-2…and Enteroviruses in Sewage Water” would be more accurate.

We changed the title into “Identification of SARS -CoV-2 and Enteroviruses in the Sewage Water – A pilot study”, as suggested.

Line 14: “All samples…”. The authors must be clear that the 100% rapid was probably due to cross-reactivity. These results are false positives.

We made the statement clear: “meaning either that antigenical structure was present in all samples, but the RNA of SARS-CoV-2 was not detected in 50% of samples, or that the rapid antigenic SARS-CoV-2 tests were cross-reactive with another proteic structures present in the sewage waters, and the results could be considered as false-positives. This last explanation is even more likely, considering the 100% positivity rate, and the fact that the test has no antigenic control.”

Line 86 -89: It is not clear what is meant by “100 ul of sewage water concentrated with 300 ul…”

We changed to: “diluted 1:4 (100 ml concentrated sewage water with 300ml…”

Line 96-98: two points here. 1) Why are these words bolded? 2) Here is the issue previously brought up. Sensitivity and specificity is for respiratory samples not environmental samples from two phase. These numbers are misleading.

  • Now there is nothing bolded anymore.
  • We changed to: “The panel tests for 19 viruses and 4 bacteria which cause respiratory tract infections, with an overall sensitivity and specificity of 97.4% and 99.4% respectively, for the nasopharyngeal samples”.

Discussion: Limitations are fine, but I think the authors should try to emphasize the false positives in the antigen tests. This is a serious limitation and authors should be very forthcoming with this information. Researchers would want to know that antigen tests for sewage samples are not a good test strategy.

We made the changes, emphasizing the idea of false positives:

“We made the statement clear: “meaning either that antigenical structure was present in all samples, but the RNA of SARS-CoV-2 was not detected in 50% of samples, or that the rapid antigenic SARS-CoV-2 tests were cross-reactive with another proteic structures present in the sewage waters, and the results could be considered as false-positives. This last explanation is even more likely, considering the 100% positivity rate, and the fact that the test has no antigenic control.”